# Validation of a Novel Stereo Vibrometry Technique for Spiderweb Signal Analysis

**DOI:** 10.3390/insects13040310

**Published:** 2022-03-22

**Authors:** Nathan Justus, Rodrigo Krugner, Ross L. Hatton

**Affiliations:** 1Laboratory for Robotics and Applied Mechanics, School of Mechanical Industrial and Manufacturing Engineering, Oregon State University, 204 Rogers Hall, Corvallis, OR 97331, USA; ross.hatton@oregonstate.edu; 2United States Department of Agriculture, Agricultural Research Service, 9611 S. Riverbend Ave, Parlier, CA 93648, USA; rodrigo.krugner@usda.gov

**Keywords:** spiderweb vibrometry, video vibrometry, black widow

## Abstract

**Simple Summary:**

Spiders often use their webs as sensory mechanisms, obtaining from them such information as the location of prey, the presence of rival spiders, and the characteristics of potential mates. Examining how this information is transmitted through the web and received by spiders is a promising biological area of research that could provide insight into a spider’s world and lead to new technologies that leverage these discoveries. In this paper, we develop a novel noncontact technique using two video cameras that is capable of analyzing vibrational signals transmitted through spiderwebs and validate this technique against the current standard of laser Doppler vibrometry. By combining the principles of stereo vision and video vibrometry, we can automatically extract three-dimensional vibrational information at any point in the spiderweb across time, and study how these signals propagate through the web. We show that this technique produces results comparable to those of standard laser vibrometry.

**Abstract:**

From courtship rituals, to prey identification, to displays of rivalry, a spider’s web vibrates with a symphony of information. Examining the modality of information being transmitted and how spiders interact with this information could lead to new understanding how spiders perceive the world around them through their webs, and new biological and engineering techniques that leverage this understanding. Spiders interact with their webs through a variety of body motions, including abdominal tremors, bounces, and limb jerks along threads of the web. These signals often create a large enough visual signature that the web vibrations can be analyzed using video vibrometry on high-speed video of the communication exchange. Using video vibrometry to examine these signals has numerous benefits over the conventional method of laser vibrometry, such as the ability to analyze three-dimensional vibrations and the ability to take measurements from anywhere in the web, including directly from the body of the spider itself. In this study, we developed a method of three-dimensional vibration analysis that combines video vibrometry with stereo vision, and verified this method against laser vibrometry on a black widow spiderweb that was experiencing rivalry signals from two female spiders.

## 1. Introduction

A spider’s web communicates a vast amount of information to its owner, in the form of vibrations that thrum through the structure. Although we are familiar with the ability of spiders to use web vibrations to identify and sense the positions of prey, intruders, and potential mates, the exact mechanisms behind how this information is transmitted remain a mystery, even for the simplest case of the planar orb web [1,2,3,4,5]. For more complex three-dimensional webs, such as those built by the western black widow (*Latrodectus hesperus*, Chamberlin and Ivie 1935), understanding how a spider might perceive these signals is an even more daunting task. To advance this quest of understanding spiderweb vibrations, we must first develop a reliable and flexible method for experimentally determining the vibrations present in a web that might be sensed by the spider. Recent advances in the field of image processing and computer vision allow for the recovery of motion signals through the analysis of high-speed video, which is the technique we used in this paper to examine and validate visual signatures of rivalry signals in the webs of our model species, *L. hesperus* [6,7,8].

The current state-of-the-art method for the vibration analysis of spider-related signals is laser Doppler vibrometry, which has commonly been used to examine the webs of *L. hesperus* [9,10,11,12,13] and both three-dimensional webs and planar orb webs built by other species [2,3,4,5,14,15]. A typical experiment utilizing a laser vibrometer involves recording vibrational data using one or more laser vibrometers aligned against the web structure or against the body of a stationary spider while the web is undergoing excitation from some signal, typically a shaker that can oscillate the web at a chosen range of frequencies. The insights gained by this spiderweb vibrometry are diverse. Previous studies have examined such things as the signal attenuation in webs for the different vibrational modes to hypothesize which propagation modalities in the web might carry the most consistently valuable information for a spider [2,3,4,10], the speed of sound and tension in webs to hypothesize how a spider might perform prey localization [4,14] or tune its web to create the most beneficial acoustic characteristics [15,16], and the vibratory characteristics of prey and mate signals that correlate to whether the owner of the web illicits a predatory or a courtship response [9,10].

Using laser vibrometry to analyze spiderwebs has a number of significant challenges: Although it is possible to align a laser vibrometer’s beam against a single strand or junction of the web, vibrations often cause the strand to leave the narrow focus of the laser, making this method of data collection unstable [4]. Additionally, *L. hesperus* are often fairly mobile during communication events, eliminating the possibility of aligning the laser against the spider itself, as has been done in former studies where the spider remained stationary in the center [2,3,4]. Previous studies using laser vibrometers have attempted to counteract this problem by aligning the laser against foreign objects such as squares of reflective tape or fly wings suspended in the web, although this has the potential to change the structure of the collected signals and can be a painstaking process [4,10,12,13]. Another limitation of laser vibrometry is the single-point nature and one-dimensionality of its data. Web vibrations can occur in transverse, longitudinal, and lateral modalities, and the type and quantity of information transmitted by each modality are still unclear [2,4,5,10]. Previous studies have responded to the one-dimensionality of laser vibrometry data by either limiting their investigation to only one of these modalities [12,13], or by performing multiple laser vibrometer recordings at different angles and relying on the planar structure of the web to inform the modality of vibrational data being recorded [2,4,5,10]. Although multiple laser vibrometers can be deployed to increase the information gained per experiment, the capability to compare vibrations across the entirety of the web and for each of the vibrational modes for the same experiment requires a new method of vibrometry.

**Figure 1 insects-13-00310-f001:**
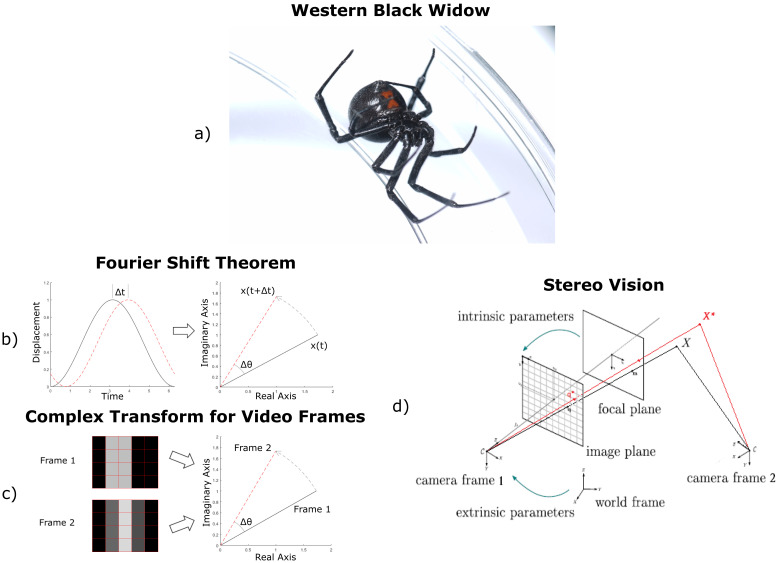
(**a**) Western black widow—*Latrodectus hesperus*. (**b**) Phase shifts from complex transforms: by the Fourier shift theorem, a spatial time-shift of a signal corresponds to a phase-shift in the complex domain. (**c**) Similarly, texture shifts in images correspond to phase-shifts in the complex domain in video vibrometry. Even if a feature moves less than a pixel, this motion registers as changes in pixel brightness values that can be used to automatically estimate motion velocity. (**d**) Pinhole camera model with augmented object position from video vibrometry in red [17]. An image is projected onto a focal plane in the camera’s coordinate frame using internal parameters, such as pixel resolution and focal distance, and then the focal plane can be placed in the world frame using external parameters, such as the camera’s position and orientation. A line is drawn from the camera focal point to the object in the focal plane. This line passes through the object in 3D space. If pixel movement values are known, this line can be shifted to get an augmented 3D position.

A high-speed video taken of the web as a whole can be used to analyze vibrations visually present in the web or spider without the need for careful alignment of the measurement instrument. Using phase-based video vibrometry on cropped subsections of this video allows for vibrometry to be performed in specific locations without the need for pixel-tracking of specific features, so long as there is sufficient degree of pixel value variation, or "texture" in the chosen analysis region. As this technique extracts information from changes in pixel brightness as texture gradually shifts from one pixel to the next, video vibrometry can give displacement resolutions as small as tiny fractions of a single pixel. The method is sensitive enough that it has been shown capable of reproducing intelligible human speech through analysis of high-framerate video of objects near a person speaking [7]. Although this noncontact technology has been previously used to examine biological phenomena such as microsaccades in the human eye [18,19] and human pulse rates through minute changes in skin color [20], it has yet to reach widespread use in the field of biology. In addition to the convenience of being able to analyze information across the entirety of a spiderweb without regard to specific sample location, video vibrometry provides vibration measurements in both the vertical and horizontal axes of the video, compared to the single depth axis of information collected using a laser [6]. This property makes it possible to apply the principles of stereo vision to video vibrometry, and combine information from two simultaneous videos of the spiderweb to analyze vibrations happening in all three spatial dimensions and at multiple points across time. This is of particular importance to black widow webs, which are highly three-dimensional mesh structures and have multiple vibrational modes in each dimension.

In this study, we developed a novel technique of three-dimensional vibration analysis by combining stereo vision and phase-based video vibrometry, and then applied this technique to extract three-dimensional vibration information from a black widow spiderweb during female–female displays of rivalry. We first describe the technique in detail, and then verify this method by comparing results from stereo vibrometry with information extracted with a laser vibrometer from a paper cube suspended in the web. We then discuss information that we can gain from stereo vibrometry that would be difficult to collect with a laser.

## 2. Background

This work combines two areas of research: video vibrometry and stereo vision.

### 2.1. Video Vibrometry

Video vibrometry extracts approximations of pixel velocity values from shifts in pixel brightness values between two adjacent video frames in regions of large pixel value variation without the need for the selection and tracking of specific features. Repeating this process for sets of adjacent frames in the video gives estimates of local motion in the horizontal and vertical coordinates of the video over the length of the examined time. The principles of phase-based video vibrometry lie in the application of the complex-valued pyramid transform, which is constructed by repeatedly applying a complex filter across multiple orientations and spatial scales [6]. This transform moves information from the real-real time domain to the real-imaginary complex domain, which has been shown to be a more reliable format for signal analysis [6,21,22]. Just as a spatially translated signal results in a change in phase in the complex domain through the Fourier shift theorem, spatial translation of pixel intensity values in subsequent frames of a video result in changes in phase through the complex-valued pyramid transform. This concept is illustrated in Figure 1b,c. Complex filters have been used for many computer vision tasks, including image orientation analysis and edge detection [22]. These pyramids are constructed for consecutive frames, and phase differences in these pyramids correspond to spatial shifts in texture.

### 2.2. Stereo Vision

Two images from different cameras can be combined by leveraging the pinhole camera model, shown in Figure 1d [23]. First, the image is projected from the image plane onto a focal plane in the camera coordinate frame at a fixed distance in front of the origin. This transformation of the image is performed using the camera’s intrinsic properties: The resolution of the image, the location of the center pixel, and the focal length of the camera, which determines the millimeters per pixel in the focal plane. The focal plane projection is then moved to the world frame using the camera’s extrinsic properties: how the camera is positioned in the world frame and the pan/tilt rotation of the camera. The line connecting the camera’s focal point to the representation of an object in the transformed image in the camera’s focal plane will pass through that object in 3D space. This process is repeated for the second camera, and the intersection of these two lines is the approximate position of that object in 3D space.

## 3. Materials and Methods

To conduct our experiment, we first positioned a transparent box containing a female *L. hesperus* specimen and web on a vibration isolation table, along with two cameras and a laser vibrometer. A second female was placed inside the box on the web, and data from the ensuing confrontation were captured from the three instruments. Calibration of the cameras was conducted using approximate measurements taken of their respective positions relative to the spiderweb, and then refined using known measurements from the videos. The data from the communication exchange of web-jerk events from the web owner captured with the cameras were then processed using video vibrometry.

### 3.1. Experimental Setup

For the experiment, two Chronos 1.4 high-speed monochrome cameras (Kron Technologies Inc., Burnaby, BC, Canada) were rigidly attached to a vibration isolation table such that the only possible allowed movement of the cameras relative to the table were pan and tilt rotations. These cameras were both connected to the same trigger, which signalled capture of the synchronized videos. The cameras were set to record 16 s of video filmed at a framerate of 1000 fps and a resolution of 600 by 800 pixels.

In the center of the vibration isolation table was placed a 30 cm transparent cube in which a female *L. hesperus* had constructed a web. On the front of the cage we placed a 2.54 cm square grid for use in camera calibration. The web was lit from above to provide good contrast of spiderweb features against the dark cage background. Measurements were taken with a tape measure to determine the horizontal position of each camera with respect to the front face of the cage. The maximum angle of the cameras with respect to each other was constrained by visibility into the cage, which was opaque on the sides.

A PDV-100 laser vibrometer (Polytec Inc., Irvine, CA, USA) was positioned between the cameras, and measurements were taken to determine its position relative to them. Days before the experiment, small paper cubes were sprinkled into the spider’s web, and the spider was given time to cut down some of the cubes. The laser vibrometer was aligned against one of the remaining cubes.

An “intruder” *L. hesperus* was placed on the web while the spider who built the web was in the retreat. The laser vibrometer was set to record the entire interaction, and the cameras captured 16-second bursts of actions whenever the experiment operator determined a communication exchange was occurring. Video pairs with large amounts of activity were saved and later used for calibration and analysis.

### 3.2. Camera Calibration

Camera calibration can be performed using any software that minimizes error in predictions for camera pose, camera focal length, and camera pixel size using given calibration points in each video. The specific methodology employed for camera calibration in this work is described in this section.

The world coordinate frame axis directions were defined relative to the front face of the spider cage. Positive X pointed to the right on the front face, positive Y pointed up, and positive Z pointed towards the cameras away from the cage. The positions of the cameras in the XZ plane were assumed known from the measurements taken during experimental setup, and the resolution of the video and location of the center pixels used for the intrinsic transformation were known from the camera settings. The remaining unknown parameters were the pan and tilt angles of each camera, the Y translation of the cameras relative to the spider cage, and the focal distances of the cameras.

To eliminate the need for knowledge of camera Y displacement relative to the vibration isolation table, the origin was chosen as the point halfway between the walls of the cage lying on the horizontal epipolar line of the video. The horizontal epipolar line was found by locating the intersection of the eight lines in each video that are parallel to the XZ plane (the top and bottom of each cork triplet, and top and bottom of each horizontal slit in the walls of the cage, seen in Figure 2). As the only allowed rotations of each camera are pan and tilt, the horizontal epipolar line represents the XZ plane that contains the focal points for both cameras, and Y translation can be considered zero.

To calculate the remaining unknowns of pan angle, tilt angle, and focal distance, calibration points were selected that were known to lie on the XY plane. As the dimensions of the cage were known, the points on the horizontal epipolar line that intersect with the cage walls have known positions. For each 2.54 cm square grid taped to the front of the cage, the four corners have known distances relative to each other.

An initial guess for the pan angle, tilt angle, and the focal distance were taken, and the discretized parameter-space around this point was exhaustively searched for the configuration that minimized the summed squared error of the origin, the cage wall points, and the distances between the points on the calibration squares for each transformation. Resulting errors between the known locations of the calibration points and the estimated positions of calibration points projected onto the front of the cage using the minimum-error configuration were all less than half of a millimeter. This minimum-error prediction of the pan/tilt angles and the focal distance along with the known camera translation and resolution comprised the final transforms used to analyze stereo data of the spiderwebs.

### 3.3. Stereo Analysis

In order to extract the stereo vibration information from the videos, each video was first cropped down to the region to be analyzed. This was done for both the resident and intruder spider and for the paper cube that the laser was aligned against.

Complex-steerable pyramids were constructed for each cropped video using the complex filter taps outlined in [22], which have been used previously for similar work [6,21]. For each frame of the video, the layers of the pyramid are constructed by downsampling the base image of the previous layer by a factor of two, then applying a complex filter pair to the downsampled base image for both the horizontal and vertical direction in the video. “Phase changes”, which correspond to shifts in pixel intensity between two frames as objects move relative to the camera, were extracted from the pyramids for each pixel across both videos and the phase signals were denoised: first spatially using an amplitude-weighted blur to squash noise in textureless regions of the video, and then temporally using butterworth mid-pass filters built to pass signals from 5 to 100 Hz. The denoised phase velocity signals were integrated to get displacement estimates for each pixel, and the displacement signals were averaged over the entirety of the cropped region to give local motion estimates in camera frame X and Y in units of pixels for each region of interest. These video vibrometry operations were performed with the aid of a Matlab GUI written for this purpose, available on github https://github.com/NathanJustus/VideoVibrometry_MatlabApp (accessed on 15 March 2022).

Stereo data composition was achieved by first finding initial pixel coordinates representing the spider or cube being analyzed in the first frame of each video. This task was performed by hand for each camera. These points were then projected onto that camera’s focal plane in the world coordinate frame using the intrinsic and extrinsic transformation parameters found during camera calibration, and constructing the line connecting that camera’s focal point to the point to be examined in the focal plane. The point that minimized the distance to the corresponding line from each camera served as our guess for the object’s initial position in space.

The initial pixel position of the object in each video was then augmented using the pixel displacement estimates calculated using video vibrometry, and the transformation process was repeated to obtain the new predicted position. Carried out for both videos, this produced a 3D local motion estimate across time for the region being examined.

Finally, for comparison with the laser vibrometer, the 3D displacement data were projected onto the line connecting the predicted coordinates of the paper cube the laser was aligned against to the measured position of the laser vibrometer relative to the cameras. Stereo data were aligned to the laser data manually by examining the time signatures of major signalling events.

## 4. Results

To examine the results of this experiment, we first validate measurements of the signals created by the *L. hesperus* and captured with stereo vibrometry against the measurements of the signals captured with laser vibrometry. Once our confidence in the method is confirmed, we examine data collected with stereo vibrometry for insights that cannot be attained through similar measurements taken with laser vibrometry.

### 4.1. Signal Pattern Analysis of Paper Box

From the laser vibrometry web displacement estimates illustrated in the top portion of Figure 3a, it is clear that the signals were composed of two sets of three individual events. The first three events (from 0 to 5 s) are visible only in the laser vibrometry signal and are much smaller. These events lie within the noise of the stereo data from the paper box. For the laser, the noise in regions with no recognizable signal is around 1 mm, and the noise of the stereo method in the same region is approximately 2 mm. Most of the noise present in the stereo vibrometry data is likely due to stereo calibration error and to the fact that the vibrations were projected onto the axis of laser vibrometer measurements, which is one of the hardest axes to measure. As motion in this axis tends to move objects towards and away from each camera rather than side to side or up and down, this motion is fairly difficult to detect using stereo vision. The raw video vibrometry displacement data have an ambient noise of approximately one hundredth of a pixel, making it much less noisy to measure signals in other axes. This noise could likely be improved by mounting the cameras further apart from each other or by using better software specifically designed for the general problem of camera calibration for stereo vision.

The latter three signals (from 6 to 16 s) were more intense and were picked up fairly equally by both stereo and laser vibrometry. Both methods estimate peak displacement amplitudes of 20 to 30 mm, but the stereo vibrometry signal prediction decays slightly faster than the laser vibrometry prediction. These are rather large-amplitude vibrations. In general, stereo vibrometry will be most useful when analyzing motions that are large enough to cause significant changes in pixel brightness (likely on the order of a mm or so in the case of spiderwebs depending on camera choice and experimental setup) but are also small enough that they do not cause relevant objects to leave the frame of the video.

It is also worthy of note that the laser vibrometer was oriented approximately with the Z axis. The cameras each have a rotation angle of approximately 15 degrees with respect to the Z axis, giving them individually very little information about vibrations in this axis. However, by combining their information using this stereo technique, vibrations in this most difficult axis to measure can produce results comparable to those of a laser vibrometer.

### 4.2. Signal Power Analysis of Paper Box

Both the laser vibrometry data and stereo vibrometry data were further analyzed by applying a filter that estimates time-domain signal power. For stereo vibrometry, this analysis was performed on the total 3D displacement magnitude predicted of the paper cube. The results are illustrated in Figure 3b.

The time-domain power estimate comparison with laser vibrometry shows that the 3D displacement magnitude from stereo vision correctly estimates times of peak power. This verifies our confidence that stereo vibrometry supplies accurate measurements of signals in the web. This time domain analysis of stereo vibrometry data is also a technique that could be useful for future work analyzing the *L. hesperus* rivalry displays, as it would be feasible to use this result to automate the extraction of large black widow rivalry signal patterns.

### 4.3. Stereo Vibrometry Analysis of Intruder Spider

Having verified the stereo vibrometry technique against laser vibrometry, we can use this process to take measurements not possible with the laser vibrometer. When analyzing spider rivalry signals, it will likely be most important to measure the signal felt by the spiders themselves. This measurement cannot be taken with the laser, as the spider shifts around in the web during the rivalry displays, and the laser must be aligned against a stationary point. Video vibrometry, however, allows these signals to be measured during times in the video where the spider remains stationary, even if it has shifted after camera setup. The y-axis stereo vibration signal for the intruder spider is shown in Figure 3c.

In the displacement signal from the intruder spider, the smaller signals that were hidden in the paper cube stereo data are now clearly visible. It is also possible to see a fourth small vibration event between two larger signals at 13 s. This event was hidden by noise in both the laser vibrometry and stereo vibrometry data from the paper cube in Figure 3a. This hidden signal found with stereo vibrometry shows that using vibration signals taken directly from the black widow spiders with stereo vibrometry as opposed to vibration signals from paper cubes sprinkled in the web can reveal important rivalry signalling patterns that would otherwise be missed using conventional techniques. This measurement is only possible using video vibrometry.

As the high mass of the intruder spider relative to the paper cube causes it to vibrate at a lower frequency with a cleaner signal, the vibration of the spider is much easier to read with stereo vibrometry than the signal from the paper cube. Additionally, the texture in the cropped region is dominated by the intruder spider itself rather than individual strands of the web, so the resulting signal has much less noise.

As noted earlier, because the cameras are positioned at an orientation of approximately 15 degrees with respect to the Z axis, reconstructing vibrations aligned with the axis of the laser vibrometer produces fairly noisy results. However, by choosing to analyze the intruder spider vibrations in the Y direction, to which both cameras can capture the full range of motion, the two camera vibration signals act as multiple sensors reading the same motion, producing lower levels of noise in the combined result. This aligns with our understanding of binocular vision in general: it is more difficult to perceive motion coming towards or away from the viewer than it is to perceive motion going side-to-side or up-and-down.

### 4.4. Web Signal Wave Speed Estimation

Rather than analyzing individual points in space, we can also use stereo vibrometry to characterize the behavior of the web as a whole. For instance, we can use this technique to estimate the wave speed of the signal as it travels through the web. By examining the time difference between the first peak displacements of the three large signal events at the origin of the signal and at the laser-aligned paper cube, we can approximate the time it takes for the signal to travel across the web. By combining this information with our knowledge of the three-dimensional positions of these points in space from stereo vision, we can calculate the wave speed of the signal as it travels through the web. An illustration of this process can be seen in Figure 3d. For the three large signals in the video, the calculated wave speeds were 24.2 m/s, 17.7 m/s, and 16.6 m/s, for an average predicted wave speed of 19.5 m/s. This estimate is fairly low compared to previous attempts to measure low-frequency transverse wave speeds using multiple sensors on the web, which have given results of 67.6 m/s [4] and 109 m/s [14]. Both of these measurements were performed on planar orb webs, so it is unclear how the speed of sound would be affected by a web with a three-dimensional mesh structure. Such measurements have been used in the past to make predictions about how a spider might perform prey localization [4,14], gather information using signal lines while away from the center of the web [15,16], and alter the structure of the web to create better acoustic properties [5,16].

## 5. Discussion and Future Applications

There are a few considerations that must be kept in mind when deciding to use stereo vibrometry to examine vibrations in spiderwebs or other three-dimensional structures. The first is that the frequency bandwidth of this method depends on the framerate of the camera being used to record the vibrations. The Nyquist frequency (one half of the camera framerate) determines the absolute maximum possible vibration frequency that can be recorded. However, this upper bound can prove quite noisy, and it is more typical to limit analysis to a quarter of the sampling frequency: a 1000 frames-per second recording would typically only be used for frequency analysis up to around 250 Hz, whereas laser vibrometry has a very high sample rate and is typically rated to measure signals up to 2 MHz [24].

Another consideration that must be thought through before employing this technique is that of vibrational signal noise, which will depend on experimental design and the vibrational axis chosen to examine. Although the resolution of phase-based video vibrometry has been shown to be accurate down to a few thousandths of a pixel [6], it is still of much lower resolution than a laser vibrometer, which is rated on the order of nanometers [24]. Performing stereo vision also increases the measurement noise over that of two-dimensional video vibrometry because of camera calibration error and binocular vision effects. A camera can only effectively measure motion that moves features horizontally or vertically in the image plane, and so vibrational information from stereo vibrometry will have noise dependent on the desired measurement axis. In Figure 3a, the stereo vibrations were projected onto the axis of the laser vibrometer, one of the noisiest possible axes, giving a noise of around 2 mm. However, in Figure 3c a better axis is chosen, giving a noise of around 1/3 of a mm.

In general, laser vibrometry will be more specialized at reading vibrations in a single dimension at an individual focus point, giving better frequency ranges and sensitivity, whereas stereo vibrometry allows for more general sample collection throughout the entirety of the web and in all dimensions at the cost of being limited by the frame-rate of the camera and the minimum resolution of a pixel allowed by camera focus and placement of the camera relative to the subject material. When considering experimental design, a camera with maximum feasible resolution should be placed as close as possible to the spiderweb in order to maximize pixel density throughout the web, while simultaneously ensuring that the camera is far enough away that proper focus can be achieved and that the image is sharp. When considering noise in each vibrational axis, the orientation of the two cameras with respect to each other plays an important role. Ideally, the two cameras should be orthogonal. However, this is often not possible because of spider cage design, so it should be acknowledged that when the cameras are closely aligned, it will be more difficult to detect motion data in the direction going towards or away from the camera lenses.

Despite the considerations that must be employed for this technique, the benefits are immense. This technique does not rely on pixel tracking of features but rather autonomous detection of pixel intensity variations between camera frames, allowing for resolutions much lower than a single pixel [7]. As stereo vibrometry gives 3D displacement data, the technique allows for the simultaneous analysis of all of the longitudinal, transverse, and lateral vibration modes along the spiderweb, presuming that these axes of vibration can be intelligently chosen. Another convenience is that video vibrometry can be performed on any subsection of the video that has adequate focus [6,7,8]. This means that stereo vibrometry allows for simultaneous sampling across the entirety of the spiderweb, allowing for simple investigations into spiderweb characteristics such as signal propagation speed and attenuation rates for each of the different vibrational modalities. Although this study focused specifically on validating stereo vibrometry against laser vibrometry and performing vibrometry on the spider bodies themselves, potential future work could perform vibrometry on nodes in the web mesh itself, giving insights into how spider signals change as they propagate through the web.

For our particular model species of *L. hesperus*, we hope that this technique can provide insights into female–female rivalry signals, which we hypothesize could lead to novel chemical-free signal mimicry techniques in arachnid dispersion that minimize the chances of black widows being collected with table grape clusters during harvest. Similar research into vibrational mating signals of another grapevine pest, the glassy-winged sharpshooter *Homalodisca vitripennis* [25], has shown that these signals can be exploited and mimicked to cause disruptions in mating behavior without the application of chemical pesticides [26]. Due to export concerns, pesticides are currently deployed to minimize accidental widow collection during grape harvest, and non-chemical forms of arachnid dispersion could alleviate undesirable environmental impacts of pesticides that are deployed to kill otherwise beneficial inhabitants of the grapevine ecosystem [27]. Other uses of this technique for spiderwebs include examinations of courtship signals and discovering how spiders might locate their prey.

Although this paper specifically investigates spiderweb signals, we suspect that this technique is also applicable to other fields of biology that are typically inaccessible to study through laser vibrometry, such as the communication of bees and wasps, wing beat patterns of hummingbirds, and even vibrations in thin plant membranes. Any source of vibration that can be captured with a high-speed camera may become subject to biological analysis.

## 6. Conclusions

In this paper, we described an efficient method for measuring 3D motion for vibration analysis by combining the concepts of stereo vision and video vibrometry. We extracted the local phases from two videos using video vibrometry, used this phase to estimate local motion over time in the camera frame, and then used these motion estimates in a stereo vision model to augment estimates of the 3D position of objects of interest. We then implemented this method to analyze vibrations in female–female *Latrodectus hesperus* displays of rivalry and demonstrated that stereo vibrometry produces results comparable to those of a laser vibrometer. We also discussed measurements that can be taken from the spiderweb using stereo vibrometry that are difficult to achieve using laser vibrometry, such as analyzing vibrations felt by the *L. hesperus* themselves and estimating the wave speed of rivalry signals through the web. We think that this technique can make spiderweb vibrometry more convenient by reducing the experimental burden of laser vibrometry, and that it opens the door to new studies into how signals propagate through the structure of a spiderweb.

## Figures and Tables

**Figure 2 insects-13-00310-f002:**
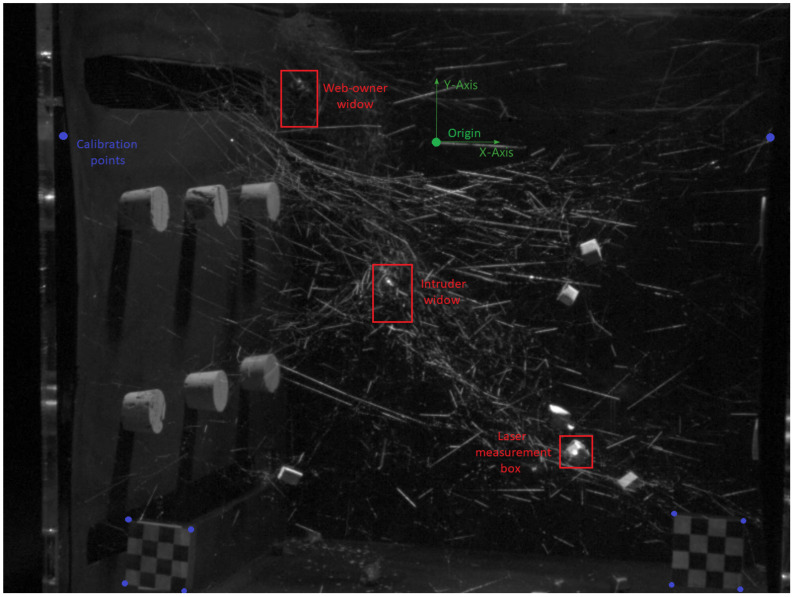
A frame from the right camera video used in the calibration process: The green point represents the chosen origin along with coordinate axes, and blue points represent coordinates with known distances to the origin or between each other. Regions highlighted in red were cropped in order to obtain local motion signals for that area of the web. A strength of the video vibrometry technique is that although the spiders are barely visible in this frame, local variations in pixel intensity over the course of the video are sufficient to predict vibrational motion for the spiders without the need for pixel tracking of specific spider features.

**Figure 3 insects-13-00310-f003:**
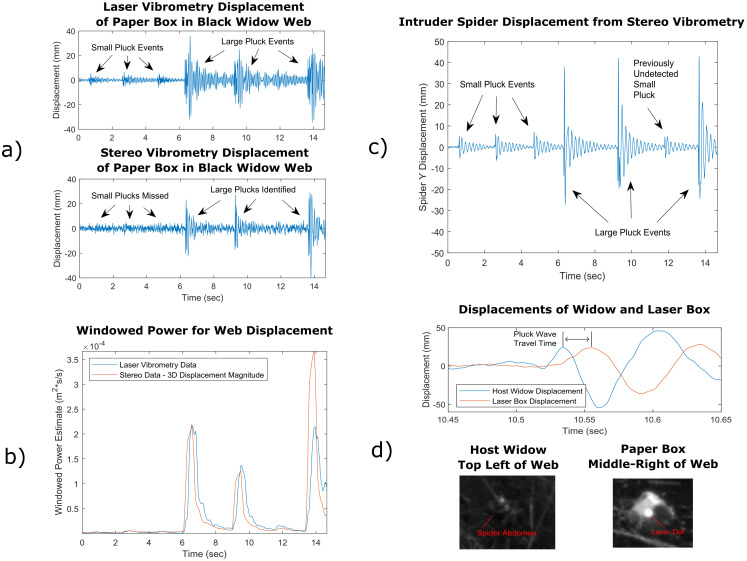
(**a**) Stereo vibrometry comparison with laser vibrometry: The stereo vibrometry data of a paper cube suspended in the web projected to the laser vibrometry axis correctly identify three large web jerk events but miss small signals detected by the laser vibrometer. (**b**) Windowed power estimates of spiderweb signals: Time-domain power can be used to detect large signals for both laser vibrometry and stereo vibrometry. (**c**) Intruder spider displacement from stereo vibrometry: Using stereo vibrometry on a spider itself rather than a paper cube generates much cleaner signals. (**d**) Calculation of the wave speed of the web signal: Video subregions are analyzed using stereo vibrometry, and the time delay between signal peaks is used to calculate the wave travel time. Wave speed is calculated using wave travel time and 3D position estimates from stereo vision.

## Data Availability

The video and laser vibrometry files that were used to generate the results in this manuscript can be accessed online through the open science foundation https://osf.io/3xyc5/?view_only=6c3eaf6650224fd7934449bdabb75502 (accessed on 15 March 2022). The MATLAB tools written and used to perform the video vibrometry analysis can be accessed through github https://github.com/NathanJustus/VideoVibrometry_MatlabApp (accessed on 15 March 2022).

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
