# Peer review of "Validation of a Novel Stereo Vibrometry Technique for Spiderweb Signal Analysis"

_insects, 2022, doi:10.3390/insects13040310_

Round 1

Reviewer 1 Report

This is basically a methods paper that describes a technique and compares it with previous techniques. It has little in the way of biology. I am a biologist, and am poorly equipped to comment on the major points of the paper. I can only offer a few minor suggestions regarding the biology. The technique is apparently very sensitive and flexible, and an improvement over previous methods. It may in the future yield interesting biological data.

One imprecision that ought to be corrected is the use of the verb “pluck” throughout the paper and the figures to describe the movements of the spiders. “Pluck”, in the sense of plucking a guitar string or plucking a harp string (the sense implied in its use in this paper), involves pulling the string more or less perpendicular to the string, and then releasing it suddenly. Spiders seldom, if ever, pluck their webs in this way; they pull on lines more or less in the direction of the line, do not release the line. Instead they jerk the line sharply, maintaining their grip on it. This allows them to sense vibrations along that line that are caused by the jerk that they have just delivered, a point relevant to the stated objective of this paper to understand how spiders perceive their world.

I think the authors overplay the dangers of black widow spider bites while harvesting crops and would recommend that they omit the next to last sentence in the first paragraph (lines 22-24) and reference to this point in the abstract, or add references that document dangers from black widow bites during harvest.

Author Response

Please see PDF attachment

Reviewer 2 Report

This manuscript uses a combination of stereo video vibrometry and laser vibrometry to measure spider-generated vibrations when two black widow spiders are present on one web. These vibrations result in displacements large enough to be visible using high-speed video cameras, which can be quantified to generate a waveform. The authors use laser vibrometry to validate the measurements taken using video vibrometry and suggest that this may be a useful tool for measuring vibrations when laser vibrometry is not possible.

I enjoyed reading the paper and it was good to see novel methods being applied to spider web vibrations. The authors presented some appealing data to support the benefits of video vibrometry. However, the paper is currently rather inaccessible to a biological audience, with an under-developed biological context for the work, and it has no discussion section of the potential and limitations of the vibrometry approaches. I expand on these points below.

The methods section is large with some figures to support, but I’m afraid as I biologist I was not able to follow throughout what was done and why. I wonder if biologists are the right audience for this work? Otherwise, this is going to need significant simplifying so that a biologist can understand the important concepts behind the approach (for example – how was 1/100th of a pixel resolution achieved? Line 191), what its limitations and strengths are, and be able to use the method themselves.

The biological context of the findings must be expanded on significantly for inclusion in this journal – there is very little discussion of the actual behaviour observed, and what this might tell us about rivalry displays in spiders. A detailed description of the behavioural sequence observed during the rivalry display would greatly enhance the paper. The results section, for instance, does not indicate whether the observed web displacements are a result of the original spider plucking, the intruder spider plucking, or a combination of these behaviours. Indeed, how could video vibrometry be used to determine the source of the vibration? The current justification and link to ecology given in the paper is underdeveloped (lines 22-25) and I’d like to see more relevant references given in both the introduction and discussion to expand the biological context.

I think the paper would be enhanced by a discussion section outlining the limitations and potential of video vibrometry, with comparison to laser vibrometry. I’m particularly interested in differences in frequency and amplitude range, as well as their resolution. There are some points made along these lines, but it would be good for this to be summarised in a discussion and put into an expanded biological context.

Minor comments

18-21. Any references that support this hypothesis in other species of spider? A description of the observed behavioural sequence to support this hypothesis would be good to include. For example, is there any escalation between web-builder and intruder before one is ousted from the web?

22-25. Needs expanding on and to include references. The pests here are the spiders (readers might think of spiders as biological control agents rather than pests themselves)?  Need a clearer link between the current study and novel methods of pest dispersion.

31-33. It is difficult to use laser vibrometry to make direct measurements of spider silk but by no means impossible, so long as the web remains still enough to focus the laser point accurately.

Abstract and Introduction. Where possible, do avoid using the exact same phrases in each.

Introduction. Could the authors expand on previous use of video vibrometry and make the novelty and/or development of the technique clear?

Line 36. Change to ‘analyze high-amplitude vibrations present in the web’, or even better quantify the displacement range over which video vibrometry will work. It would also be good to include an idea of the frequency cap that can be analysed – which will be limited by the framerate of the camera. The limitations and scope of the technique should be clear throughout, with a dedicated Discussion section on this point.

Figure 1. The red lines on panel B make it look like air-borne sound, whereas following the silk threads would be a better visual here. Is the image of a black widow web?

Line 46. How were lighting issues dealt with to be visible at the full displacement range on both cameras?

Lines 52-53. It would be more useful to compare the benefits and limitations of both techniques in a more balanced way. This should be added to a new Discussion section.

Line 104. Please rephrase “that it found most offensive”

Figure 2. Consider adding X, Y and Z axes labels to Figure 2, as described lines 113 to 114. The one frame provided of the video footage is quite unclear – the spiders in particular are barely discernible, and it begs the question as to how these features were accurately tracked by hand. If the images were cropped for processing, perhaps insets that show these areas enlarged would be helpful in order for the reader to visualise how the tracking was carried out.

160-162. How was this carried out, exactly? What parts of the spider were used as markers?

Figure 3. Consider using either ‘laser box’ or ‘paper box’ throughout

216-249. I’m still unclear on how individual pixels on the spider and box were tracked over time (only says “performed by hand” line 161 – what parts of the box or spider were tracked between frames?). What is the estimated error from this approach? Is the method of pixel tracking consistent throughout the paper? How is rotation of the objects dealt with during tracking?

240-249. This last paragraph, particularly the last sentence, is not clear. Why is Y axis a better choice?

250-262. What are the biological implications of the estimated wave speed? How does this fit in with previous studies?

263-277. Perhaps comment on further development of the technique here – logical next step is to use video vibrometry for direct tracking of the spider web mesh (perhaps using the nodes between threads as tracking points) rather than simply objects suspended in the web?

Author Response

Please see PDF attachment

Reviewer 3 Report

This is a technical manuscript introducing a new method (video vibrometry combined with stereo vision) and comparing it with classical method of laser vibrometry. However, the title and the first half of the abstract (as well as Introduction) promise quite another topic – behaviour of a black widow. The title and abstract was misleading also for me: after reading it, I expected an ethological work. After my agreement with writing the review and reading the text afterwards, I realised that it is in fact a methodological, technical text. Let thus please accept a review from somebody, who knows biology and ethology of spiders but is aware only little about the technical background.

Title: It should clearly declare the contect of the paper. You are describing (and comparing) a new method for observation/evaluation, and the black widow is "only" your model species (not a target species). As such the name of the spider should not apper in the title – you do not give any ethological observation of this species.

Addresses and keywords: There are incomplete addresses under 1 and 3. The key words should usually not repeat the words from the title.

Abstract and Introduction: Here I have the same comment – you start with communincation between two spiders, so that the reader expect that you will deal with ethology of black widow. What I really do not understand, are the notes about "pest dispersion" (lines 4 and 23)… Yes, spiders reduce pest, but this is absoultely not related to the topic and I highly recommend omitting it from both abstract and introduction. Instead, you may provide more examples of works that had used laser vibrometry in their research (not necessary only about theridiids) and what were their limitations. From this, your own aims (the new method and comparison with "convential" method) will naturaly arise.

Background and M&M: Very nicely and precisely described. I point out only one error: The web on Fig. 1b is a 2D orb-web constructed by araneids. Showing such a web, when your model species is a black widow, would be a shame! Theridiids construct 3D "meshy" or tangle web. Please, modify this figure. And one typo: on line 147, there should be Freeman (1991) – year in parenthesis – or [9].

Results: The experiments showed clearly that stereo vibrometry 1) can provide finer detections, 2) can focuse directly on moving object (spider) and 3) allow measurements in comparison with "conventional" techniques. The only results (presented rather as a "side effect" of the experiments) dealing with the spider were the wave speeds. For this reason, I strongly recommend modification of title, abstract and Introduction of the manuscript as stated above.

Conclusion: Maybe I would rather write "Discussion and conclusion". The examples mentioned on line 277 could be explained in more details, broadly, as a discussion. I suggested above including some works to the Introduction, where your method should have been of use. Here, you can suggest, what else can be done in those concrete works (e.g. not only studying rivalry but also courtship or evaluating level of danger of the prey etc.).

References: [Note for the editors: I personally do not like when authors are asked to cite papers no older then 5 years – sometimes one must go to original sources /"ad fontes"/ often from 19th century – namely in solving "long-standing questions"]. I think that more works could be added as examples where your method would be useful (see above) but this is up to decision of the authors.

I wishou you good luck in your research :)

Round 2

Reviewer 2 Report

Thank you to the authors for your replies to my comments and edits to the manuscript. Overall, I am very supportive of the method, which I think is really useful for studying biological motion. But I still think that some edits are needed to put the work into context and make it accessible for biologists.

Comment 2.1 and 2.3. The biological context – introduction and conclusion. The introduction needs more references included, e.g. work of Masters et al., Landolfa et al., Mortimer et al., Vibert et al. on spider web vibrometry.

Comment 2.2. Accessibility of methods for a biologist audience. This is certainly better, I think more edits could be made to make this clearer and consistent. E.g. 1 explain intrinsic versus extrinsic parameters and focal and image plane in Figure 1 and Section 2.2. E.g. 2 ‘texture’ line 171 (this is explained better line 304 and figure 2 legend) and explaining line 51-52 (how fraction of a pixel) and line 73 (what is the complex-valued information). This might be more understandable if greyscale/intensity values within the box of interest are referred to. The example in Figure 1c of a light pixel moving is helpful, but the link to sub-pixel resolution is not apparent from this.

Comment 2.4 (2.13). I welcome the new discussion section. However, the discussion needs comparison to laser vibrometry, rather than just a summary of video vibrometry characteristics, to make it more balanced. For example, references to show vibrometry frequency range and displacement sensitivity/noise floor to compare to video vibrometry. References to show possibilities opened up by 3D vibrometry (to measure multiple axes of motion) or multiple laser heads (i.e. to measure propagation of signals). This allows you to emphasise the practicality and novelty of this approach (i.e. how cheap, practical, areas rather than points measured etc).

Comment 2.9 Novelty of video vibrometry. I’ve read the third paragraph of the introduction as directed, but I think the novelty of this method, and where video vibrometry has previously been applied before outside of spider webs, could be made a lot clearer. Has it been used for study of biological systems before?

Further comments.

Figure 1b. Is the Fourier shift theorem included here to explain laser vibrometry? It is only referred to in the text line 81, which makes sense for 1c. Could the relevance of 1b be explained a bit better in the text please.

Line 38 – not all studies on spider webs used reflective tape, so edits are needed to clarify (e.g. just for black widow webs?).

Line 47. The authors need to acknowledge that laser vibrometry can use a system with more than one laser. This allows detection of signals simultaneously at different locations or angles. I agree that video vibrometry will be a cheaper and useful alternative, but worth being balanced in your assessments of what is possible with laser vibrometry.

Line 101-102. Suggest including something like ‘…communication exchange (not explained in detail, but including small and large plucks from an unknown spider) captured…’ or similar.

Line 209. Noise on vibrometry signal (1mm) on the box is pretty poor. It is worth comparing this to some previously collected vibrometry data, and how the noise can be improved using laser vibrometry.

Line 218. This study looked at quite high displacements at up to 30mm. It is worth adding to the discussion what types of biological signals are likely to be within this range or higher, and which are lower. Is video vibrometry only going to be useful for displacement ranges above c. 1mm?

Line 277-281. This is possible with laser vibrometry and has been published previously. Measurement of propagation speeds through the web and characterisation of web properties can be achieved using laser vibrometry when two laser are used, e.g. Mortimer et al. 2016. Many other techniques other than light microscopy have been used to examine and quantify spider silk properties – please look at the literature and cite a broader set of references.

Lines 315-323. Please include more references in this paragraph to put this is a stronger biological context.
